



# The Pluvial Flood Index (PFI): a new instrument for evaluating flash flood hazards and facilitating real-time warning

Markus Weiler[1], Julia Krumm[2], Ingo Haag[2], Hannes Leistert[1], Max Schmit[1], Andreas Steinbrich[1], Andreas Hänsler[1]

[1] Faculty of Environment and Natural Resources, University of Freiburg, Freiburg, Germany
[2] HYDRON GmbH, Karlsruhe, Germany

*Correspondence to*: Markus Weiler (markus.weiler@hydrology.uni-freiburg.de)

**Abstract.** Pluvial (flash) floods frequently cause damage in rural and urban watersheds as a result of short-term, intense local precipitation events that cause infiltration excess runoff and overland flow. Unlike fluvial floods, pluvial floods are primarily characterized by surface runoff and flow in small ditches and creeks, making them unsuitable for evaluation using common extreme value statistics based on long-term river discharge data. Precipitation statistics alone are insufficient for predicting pluvial floods because these floods are also influenced by hydrological and hydrodynamic processes. We propose a new pluvial flood index (PFI) that considers precipitation as well as hydrological and hydrodynamic processes to assess the hazard of surface flooding. The PFI is based on pluvial flood hazard areas (PFHA), which are defined as areas where water depth, flow velocity, or both exceed thresholds that endanger pedestrians and vehicles. We defined four PFI classes based on historical and design events, ranging from no hazard to very large flood hazard. The PFI serves as a simple, dimensionless measure and information tool.

PFHA and PFI were calculated for various events using radar-based precipitation input, dynamic simulations of infiltration and saturation excess, and hydrodynamic simulations of surface runoff. PFI forecasting requires quantitative precipitation data as well as appropriate processed-based distributed hydrodynamic and hydrological models at large temporal and spatial scales. We demonstrate the PFI's applicability and utility by creating large-scale flash flood hazard maps and hindcasting an extreme historical event. Furthermore, the PFI can link to detailed local flash flood hazard information, assisting municipal decision-making. It can also be a key component in operational pluvial flood warning systems, providing information on the occurrence and severity of floods on a scale of several hectares to square kilometres. This educates stakeholders and the community, improving real-time warning systems, preparedness, and planning decisions.

## 1 Introduction

Pluvial floods are flash floods with inundation typically arising from localized, severe convective precipitation events. They are characterized by short formation durations, attributable not only to the rapid growth and movement of convective



thunderstorm cells but also to the underlying runoff generation mechanisms. Pluvial floods predominantly arise from unrestrained surface runoff caused by infiltration excess (Beven, 2004), occurring with fast moving water typically away from rivers and watercourses (Archer and Fowler 2018), even under dry preconditions. The rapid formation time and typically limited spatial extent of these events (generally only a few square kilometres) complicate prediction and real-time forecasting and result in a minimal lead time for such occurrences (Borga et al., 2011).

Due to climate change and the resultant rise in temperature, an intensification of the hydrological cycle (Huntington, 2006; Kunstmann et al., 2023) is anticipated, leading to a probable increase in heavy rainfall events (BBK 2021) and a corresponding increase in pluvial flood occurrence and magnitude (Wasko et al., 2021). The escalating threat of pluvial floods, exacerbated by ongoing urbanization and surface sealing, alongside with insufficient awareness and protective measures, suggests a significant vulnerability to damage in numerous locations, which is likely to persist in the foreseeable future. Heavy rainfall

currently constitutes around 50% of all flood-related damages in Germany (BBK 2015). A similar picture was derived for England, with 3 million properties susceptible to pluvial flooding - compared to 2.7 million properties at risk of fluvial or coastal flooding (Environmental Agency, 2018).

In addition to increasing awareness and formulating protective measures, the comprehensive categorization of pluvial flood hazards and the improvement of pluvial flood forecasting and alerts are essential components that enhance pluvial flood

management and mitigate their detrimental effects (Haag et al., 2022a). Therefore, we propose the new Pluvial Flood Index PFI as a versatile tool to reach these goals. PFI encompasses information beyond mere heavy rainfall data, as the occurrence of a flash flood from heavy rainfall is contingent upon the interplay of hydrological and hydraulic static features and dynamic processes. Unlike current fluvial flood classifications or hazard indices that primarily pertain to river flow and subsequent overbank flooding (Kazakis et al., 2015; Kabenge et al., 2017; Vojtek, 2023), the PFI is designed exclusively for pluvial events,

aiming to address the hazard arising from uncontrolled surface runoff. The PFI is characterized as a hazard index that does not account for vulnerability or damage potential, hence offering no independent assessment of flood risk. On one hand, the PFI is designed to assess and compare the susceptibility of various regions to pluvial floods. On the other hand, it is also intended as an operational index to alert the public in real-time on pluvial floods and to facilitate appropriate measures within communities. Consequently, the PFI must satisfy the following criteria (Krumm et al. 2024):

• Central to the PFI is the consideration of all hydrological and hydraulic factors and processes, in addition to precipitation, that contribute to the occurrence of local flash floods resulting from heavy rainfall events.

• The PFI should allow broad applicability and thus be founded on readily available data. The intended practical real-time application necessitates that the PFI can be ascertained rapidly while maintaining adequate precision.

• For optimal application of the PFI, the index must not be an abstract figure; it should instead be grounded in a measure
that is straightforward, comprehensible, and tangible, establishing a direct and significant correlation to the hazards associated with uncontrolled surface runoff.

• As a tool for operational public warning, the PFI must be readily comprehensible and distinctly conveyable.





We will first describe in detail the relevant processes that need to be considered when evaluating pluvial flood hazards. The definition and designation of Pluvial Flood Hazard Areas (PFHA), which form the core of the PFI, will then be explained

followed by a list of requirements for the hydrological and hydrodynamic models necessary to derive the PFHA and PFI. In the result section, we will first show the potential of the new approach with a hindcast the flash flood of 2024 in the Wieslauf catchment, Germany and then illustrating the potential of the PFI for creating a PFI or flash flood hazard map.

## 2 Defining the pluvial flood index (PFI)

### 2.1 Relevant Processes

Not every heavy rainfall event leads to a pluvial flash flood. In fact, the occurrence of a pluvial flood is contingent not only on the precipitation itself but also on several spatially and temporally variable factors and processes at the land surface (e.g., Tarboton 2003; Steinbrich et al. 2016; Ries et al. 2020). To derive a meaningful and robust PFI, it is essential to incorporate, in addition to a quantitative precipitation input, all these hydrological and hydrodynamic (hydraulic) influences and processes as depicted in Figure 1.

First, it is essential to determine the amount and intensity of overland flow generated by infiltration excess or saturation excess. The second phase occurs only when the infiltration or saturation surplus spills over the surface and accumulates to cause significant surface runoff. On impermeable surfaces, surface runoff can occur following minimal initial losses. Consequently, the ratio of impervious surfaces is a significant, spatially highly variable but temporally static factor, in the generation of surface runoff. In instances of pluvial flash floods, originating outside of settlements, the extent of impermeable surfaces is

typically small, making surface runoff from permeable areas very important. In this case, the infiltration characteristics of the soils, along with the initial soil moisture, are crucial factors. Unlike extensive fluvial floods, the generation of overland flow during localized pluvial floods is frequently more influenced by infiltration excess and the consequent Hortonian overland flow than by saturation excess form saturated areas (Steinbrich et al., 2016; Stewart et al., 2019). Infiltration excess occurs when the rainfall intensity surpasses the infiltration rate. Soil type and structure as well as the initial soil moisture strongly

influence the infiltration rate (e.g. Rawls et al, 1992; Jury & Horton, 2004). Land use and vegetation can also greatly affect infiltration, through varying macroporosity (Bachmair et al., 2009; Zhang et al., 2019, Ries et al., 2020) and varying susceptibility to siltation (Bonta & Shipitalo, 2013; Seibert & Auerswald, 2020). Soil type and land use are relatively stable temporal characteristics. However, vegetation cover fluctuates seasonally, in particular on agricultural land, and thereby influences infiltration capacity via root penetration, soil macropores and siltation in a seasonal manner (Seibert & Auerswald,

2020). Moreover, the initial soil moisture prior to an event exerts a dynamic influence on the actual infiltration capacity and possible infiltration excess (Figure 1).

Once overland flow has been generated, the terrain is crucial for the accumulation of unrestrained surface runoff and the formation of a pluvial flash flood. Attributes such as slope gradients and relief along with natural or artificial drainage



structures determine the direction and activation of surface runoff routes (Fiener et al., 2011). Surface roughness strongly influences the hydrodynamic process of flow accumulation. Therefore, different spatially and temporally variable surface characteristics, that typically influence surface roughness, have to be taken into account (Fiener et al., 2011; Seibert & Auerswald, 2020). This is particularly important for surface runoff, where roughness coefficients are also influenced by flow depth and small micro flow paths (Oberle et al., 2021) Frequently, erosion and sedimentation by surface runoff goes hand in

hand with larger pluvial flood events (Figure 1), in particular on low-permeable surfaces and areas with low vegetation cover (agricultural, burned areas and badlands) (Lange et al., 2003; García-Ruiz et al., 2008). Both geomorphological processes may alter the flow pathways, trigger surface retention and significantly alter the impact of surface runoff on infrastructure and buildings, but are typically not even considered in most hydrodynamic flood models (Hamidifar et al., 2024).

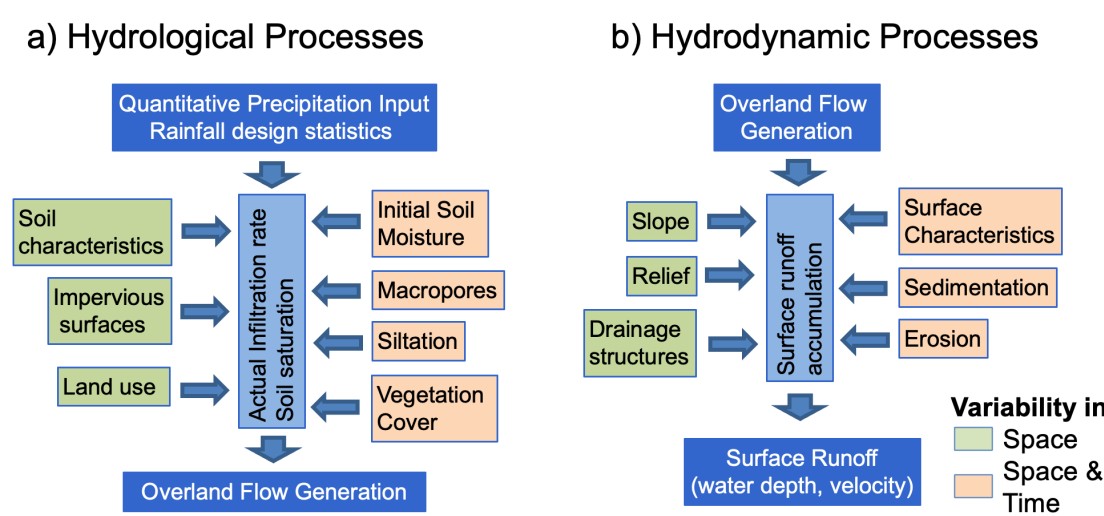


**Figure 1: Factors and processes specific to the generation of overland flow and resulting surface runoff due to a) hydrological processes and b) hydrodynamic processes.**

**2.2 Pluvial Flood Hazard Areas (PFHA)**

The PFI refers to the area-wide hazard source of unrestrained surface runoff. This hazard can best be quantified by the extent of the areas where there is a thread from surface runoff and flooding. A hazard is defined as either pedestrian no longer being able to safely cross a flooded area or the inability of vehicles to safely navigate. The floating of vehicles naturally poses a significant hazard to people, namely the vehicles occupants and pedestrians, who may be affected by the drifting vehicles. The

areas where such a hazard to pedestrians or vehicles exists are hereinafter referred to as *Pluvial Flood Hazard Areas* (PFHA).




The criteria for defining the PFHA can be obtained from current research on the stability of pedestrians and vehicles at specific flow velocities and water depths. Figure 2 presents the summary evaluations on the stability of pedestrians (a) and vehicles (b) as reported by Martínez-Gomariz et al. (2016; 2018). This leads to the conclusion that a hazard for pedestrians is fundamentally characterized by the product of flow velocity (v) and water depths (z), represented as a specific surface runoff (q = v * z). At elevated flow velocities, the flooding depths becomes insignificant; thus, a hazard exists even at low water depth purely owing to the flow velocity. For vehicles, a hazard is encountered only with a greater specific discharge than for pedestrians. Nevertheless, lightweight compact vehicles can already remain buoyant at relatively shallow water depths, irrespective of the flow velocity. To provide a conservative, or safe, delineation of the PFHA, which covers the potential risk to elderly individuals or lightweight compact vehicles, a comprehensive envelope of all three hazard factors was employed. The threshold for pedestrian hazard based purely on flow velocity, irrespective of water depth, was set to 1.5 m/s, as also recommended by practitioners, which is considerably more stringent than the guidelines proposed by Martínez-Gomariz et al. (2016). The region characterized by elevated flow velocities, termed the "high hazard zone" for pedestrians by Martínez-Gomariz et al. (2016) (refer to the yellow box in Figure 2a), is deemed hazardous irrespective of the water level. The threshold for water depth to ensure no hazard in cars was set to 0.3 m.

## a) Pedestrians

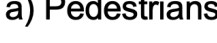
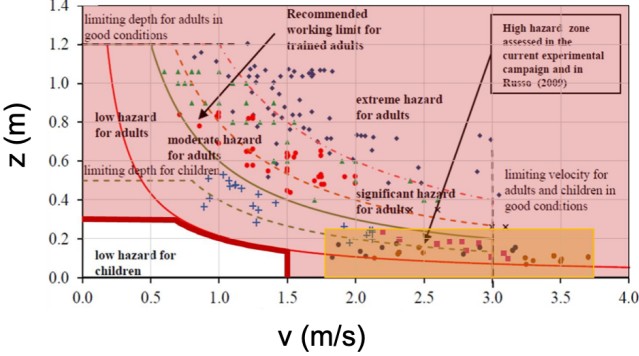

## b) Vehicles

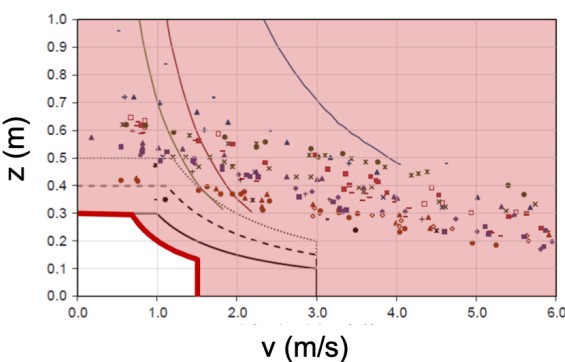

**Figure 2: Experimentally derived criteria (water depth z and surface runoff velocity v) for pedestrians (a) and vehicles (b) (Source: Martínez-Gomariz et al. 2016; 2018) supplemented by the selected criteria for delineating pluvial flood hazard areas (red thick line) and the conditions of PFHA (red transparent area).**

The criteria for delineating the pluvial flood hazard area (PFHA) are satisfied when the following conditions occur:

$$PFHA = \left\{ v \geq 1.5 \ \tfrac{m}{s} \right\} \cup \left\{ z \geq 0.3 \ m \right\} \cup \left\{ q \geq 0.2 \ \tfrac{m^3}{s*m} \right\} \tag{1}$$

Hence, PFHA are areas where pedestrians or vehicles are at risk as water depth, flow velocity or the combination of both exceed the defined thresholds.



### 2.3 Pluvial Flood Index (PFI)

Pluvial floods are rare, typically local events, complicating the establishment of dependable, site-specific return intervals as established for fluvial floods. Moreover, unrestrained surface runoff or the resultant PFHA is not a viable operational measurement. Hence, long-term time series of the extend of PFHA or other pluvial flood metrics are not existing. Consequently, return periods of pluvial floods must be determined using model calculations, leading to significant uncertainty of the return period. Moreover, the application of return periods (e.g., Grisa 2013) and its communication with the public may

lead to misunderstandings. Finally, no protective framework for pluvial floods exists, in contrast to fluvial floods, which are usually associated with a specified return period to ensure specific protection measures (e.g. protection against a flood with a return period of 100 years: HQ100).

    To capture the hazards of a flash floods, the absolute extent of the PFHA is therefore more crucial than the occurrence probability of an event. Absolute thresholds also provide comparability among different regions (e.g., with the assumption of

identical precipitation and initial soil moisture conditions) and allow to map the PFI hazard. The pluvial flood index (PFI) is fundamentally based on the relative area fraction of the pluvial flood hazard area (PFHA) to a reference area. The PFI thus represents the relative fraction of the total area where there is a hazard to pedestrians or vehicles due to uncontrolled surface runoff and flooding. Thus, the PFI has a comprehensible, clearly interpretable basis that has a tangible reference to the main hazard source of flash floods.

In addition to defining the PFHA, the reference area must be determined for which the fraction of the PFHA is calculated. As reference areas (or denominator), catchment areas (e.g., basic catchment areas), uniform grids, or buffered regions (e.g. circles) are fundamentally suitable. Additionally, it would be conceivable to determine the PFI only for vulnerable areas of a catchment, for example, only the PFHA fraction in settlements, roads or densely built-up areas.

    The PFHA fraction and hence the PFI strongly depends on the size of the reference areas. Natural catchment areas, and

especially intermediate catchment areas, are very heterogeneous in terms of their size distribution. In addition, radar products of precipitation, which can be used as input for simulating PFHA and PFI, are usually available as raster data and would need to be regionalized again for the catchment areas when using catchment areas as reference surfaces. Using uniform grids as the reference area for determining the PFI could be an option, but the gridding extend and boundaries among the grids may have strong effects on the PFI. Therefore, we propose instead of using a uniform, regular grid (as typically done for determining

heavy rainfall from radar data), to use a moving circular buffer (uniform or weighted by the radius) with the recommended area to visualize the PFI in space.



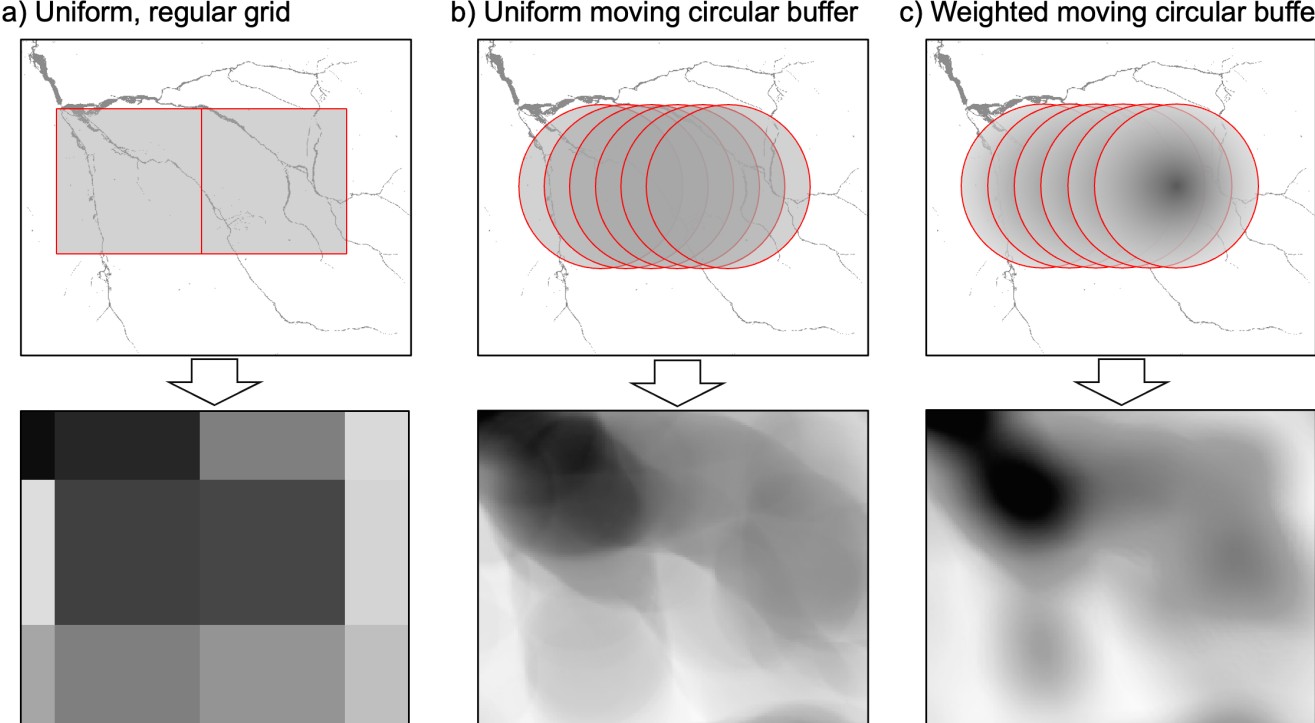

**Figure 3: Comparison of the three potential approaches to derive the PFHA fraction from the PFHA for a defined reference area with a) shows the example of applying a uniform, regular grid, b) a uniform moving circular buffer and c) a weighted (radial linear) moving circular buffer with the resulting map of PFHA fraction with lower brightness indicating higher fractions.**

The larger the reference areas are, the more the local maxima of the PFHA fraction are dampened and the more the differences among neighbouring areas are blurred. In addition, for the determination of the PFI, depending on the approach, it may be necessary to assume a comparable amount and intensity of precipitation for the area to be evaluated. In the case of local heavy rainfall, this area should generally not be larger than approximately 5-10 km² (Lengfeld et al., 2019). On the other hand, disproportionately small reference areas lead to local peculiarities being overemphasized and uncertainties in determining the PFHA resulting in significant uncertainties in the reference area-specific PFI. Furthermore, the PFI is intended for a broad assessment of flash flood hazard, for example at the state level, where regional rather than local differences are to be captured. Against the outlined background and based on experiences with heavy rainfall hazard maps and different evaluations in many regions and for many past events, we recommend an area of 2 to 4 km² as the reference area and we will use 2 km² in for generating our results.

In theory, the PFHA fraction can range between 0 and 1 (or between 0% and 100%). The PFI is intended to serve as a dimensionless index, primarily for simple and clear communication with the general public. For the information and communication with specialized users, an index is, however, typically only partially suitable. To ensure that the PFI serves this user group optimally, the PFI can additionally be linkable to more detailed specialized information like detailed hazard maps of PFHA of a settlement.





Absolute thresholds are favoured in the context of the PFI for the reasons already stated. These are uniformly established across regions, ensuring comparability among them. In accordance with various stakeholders form the government and flood forecasting services in Germany, the PFI was categorized into four classes according to three thresholds (Table 1). This

quantity of classes communicates more than a just binary information (flash flood hazard: yes/no). Conversely, the quantity of classes stays reasonable, allowing for the assignment of a descriptive significance to each class and facilitating unambiguous differentiation among them. A considerably greater number of classes may indicate a classification accuracy that does not align with the actual precision attainable on a broad scale by the PFHA. There are no clearly objective criteria for establishing the thresholds for PFI classification. Nevertheless, the thresholds of PFHA fractions were determined as accurately as possible

in order to capture relevant hazard situations. However, the thresholds can be adopted to different regions and countries and the proposed levels only serve as an initial definition for different pluvial hazards in Germany.

**Table 2: Classification of the PFI into different absolute hazard levels**

| PFHA fraction (%) | PFI | Description | Color code |
|---|---|---|---|
| < 0.5 | 0 | No hazard | #54c21f |
| 0.5 – 2.0 | 1 | Medium hazard | #FFEC01 |
| 2.0 – 5.0 | 2 | Considerable to large hazard | #E22323 |
| ≥ 5.0 | 3 | Very large hazard | #934490 |

**2.4 Selected models to derive PFHA and PFI**

In theory, a wide range of different models can be applied to derive PFHA and PFI. However, there are a few requirements for the hydrological and hydrodynamic models. The distributed hydrological models must primarily be capable of accurately representing the highly dynamic temporal formation of overland flow due to infiltration or saturation excess under high rainfall intensities. As the surface characteristic are spatially very variable, the spatial resolution should be relatively high in the order

of 1 m to 5 m to capture the size of roads and buildings. The process-based hydrological model RoGeR (Steinbrich et al. 2016; Schwemmle et al. 2024) was selected for this study as it allows long-term simulation to define initial conditions, captures all runoff generation processes at a high temporal resolution of 5 min, was evaluated in many catchments in the area of South-West Germany and even at smaller spatial scales of hillslopes with high intensity rainfall experiments (Steinbrich et al. 2016, Ries et al., 2020).

The concentration and accumulation of surface runoff can be simulated using suitable 2D hydraulic models including shallow water table calculations and an accurate mass balance. Of course, the spatially distributed and temporally dynamic input from the hydrological model must be taken into account. As we plan to derive PFHA and PFI for larger areas with a spatial resolution




smaller or equal to 5 m, computation time plays a crucial role. This is even more crucial, when the PFI is used in real time forecasting Systems. Models whose time requirements are significantly reduced even for relatively large-scale calculations compared to established models are now available (e.g. Apel et al. 2022) or simplified hydrodynamic approaches can be applied (e.g. Leistert et al., 2025). For larger scale applications, the nowadays available high-resolution DEM from airborne LiDAR sensing is in term of spatial resolution and accuracy very good (Liu, 2008; Sakensa & Merwade, 2015). However, the surface models are usually not capturing all relevant drainage structures like culverts, creeks under small bridges or other kind of artificial water retention infrastructure (Lindsay & Dhun, 2015). Hence, these DEM must be pre-processed to make sure that the derived water depths and velocity maps are as accurate as possible. In addition, the 2D hydraulic models should allow for concurrent simulation of the three variables maximum water depth, maximum flow velocity and maximum specific surface runoff (see Eq. 1). For this study, we apply the new approach AccRo (Leistert et al., 2025), as it allows for fast and reliable simulation of the necessary variables at a spatial resolution of 5 m defining the surface roughness according to the land-use and surface characteristics as defined in LUBW (2016).

## 3 Results

### 3.1 Hindcasting the flash flood of 2024 in the Wieslauf catchment, Germany

We selected an extreme pluvial flash flood event that occurred in the evening of 2nd June 2024 in the Wieslauf watershed (Rems-Murr-Kreis) to illustrate and elucidate the procedures and outcomes of our methodology for producing PFHA and PFI maps. The event caused extensive pluvial and fluvial flooding. The discharge recorded at the gauge at the confluence of the Wieslauf river into the Rems river was far above the discharge with a 100 year return period. The estimated damages in the catchment exceeded 300 million euros (Landtag von Baden-Württemberg, 2024).

The radar-derived quantitative precipitation estimates (QPE) of the event with a spatial resolution of 250 m and a temporal resolution of 5 minutes were summed up for the period between 15:30 and 21:30 and compared with observation of private weather stations (not utilized for generating the radar-based QPE) in the region and weather stations of the DWD (Fig. 3 a). The station's maximum event sum of 110-120 mm aligns well with the estimated QPE maximum of 130-140 mm. The overall quantity and geographical distribution of the region over 80 mm, as determined by radar-based QPE, were accurately represented, with minor underestimations in the northwest and southeast (Fig 3a). The convective rainfall event progressed from northeast to southwest, traversing the map extent in four hours, with two separate peaks in intensity.

Utilizing the hydrological model RoGeR, we simulated the generation of overland flow (infiltration and saturation excess) with high initial soil moisture conditions (as simulated by RoGeR's water balance module for the event date) at a spatial resolution of 5 meters and a temporal resolution of 5 minutes. Figure 3b illustrates the total overland flow generation, with elevated values in regions of peak rainfall, as well as high values associated with urban settlements, roadways, and certain soil types.







**Figure 4: Hindcasting the flash flood event in the Wieslauf catchment at 02.06.2024 with a) total event radar-based QPE including measurements at stations at reported total event rainfall, b) total overland flow, c) maximum water depth (z) of the surface runoff with d) related maximum velocity (v), e) maximum specific surface runoff (q), g) resulting extend of the PFHA, h) PFHA fraction and g) PFI using a moving 2 km² circular reference area. Spatial resolution is 5 m.**





Figure 4 also illustrates the outcomes of the 2D hydrodynamic simulation utilizing the surface runoff from the hydrological model, maintaining identical spatial and temporal resolution. The DEM was adjusted to facilitate flow in all creeks and rivers in accordance with the delineated stream network. All buildings were considered specifically in the DEM to facilitate surface runoff around buildings. The maximum water depth (Fig. 4c), maximum flow velocity (Fig. 3d), and maximum specific surface runoff (Fig. 4e) exhibit the characteristic pattern of surface runoff in valleys and concave slopes, (the underlying hillshade in

grey depicting the terrain features). The considerable rainfall event caused overland flow on the hillslope (Fig 4b), which drained into the Wieslauf River, causing it to breach its banks and generate significant fluvial flooding along the river's main stem in the lower catchment area. According to the definition of the PFHA (Eq. 1), their extent is mapped in Figure 4f. The resulting PFHA fraction (Fig. 4g) and PFI (Fig. 4h), employing a dynamic 2 km² circular reference area without an accumulation threshold, illustrates a large region with a PFI of 3 (very large hazard) and extensive areas with a PFI of 2

(considerable to large hazard), attributable to the event's extensive spatial extend. The settlements experiencing the most significant damage and fatalities were situated in the highest PFI, however considerable damage to forest roads and bridges was also observed in PFI 2.

## 3.2 PFI hazard mapping (flash flood hazard maps)

The second example illustrating the potential of the PFI is the creation of a PFI or flash flood hazard map. The objective is to

determine PFHA based on a defined amount and intensity of rainfall for a specified return period for an extensive area. In regions susceptible to pluvial flooding, the PFI should be high, indicating possible hazard when heavy rainfall events occur. The hazard maps offer an initial indication of whether towns need to implement further measures to prepare for pluvial flooding. To simulate the PFHA, we utilized the already existing overland flow generation maps created for the entire state of Baden-Württemberg to assist the "Kommunales Starkregenrisikomanagement" (LUBW, 2016).

The overland flow generation maps were produced using the RoGeR model, based on 1-hour rainfall events with varying return times (we selected a 100-year event for our example) and median initial soil moisture conditions throughout summer. The overland flow generation maps for the designated area in SW Baden-Württemberg (refer to location and extent in Fig. 4) is presented in Figure 5a. The regions exhibiting elevated levels of overland flow are situated in the western lowlands and the hilly terrains west of the Black Forest. The Black Forest itself shows low values of overland flow generation despite higher

precipitation, as the predominantly forested soils are highly permeable and runoff generation is primarily dominated by subsurface stormflow, particularly in winter (Bachmair & Weiler, 2012). Utilizing the identical 2D hydrodynamic model AccRo, now with an accumulation threshold of 10 km² to only capture pluvial flooding, we simulated the PFHA (Fig. 5b). The PFI was determined and illustrated in Figure 5c using the identical 2 km² circular reference area. The PFI identifies certain places with a high hazard, while numerous areas possess a PFI of 0, indicating no danger to pluvial flooding.






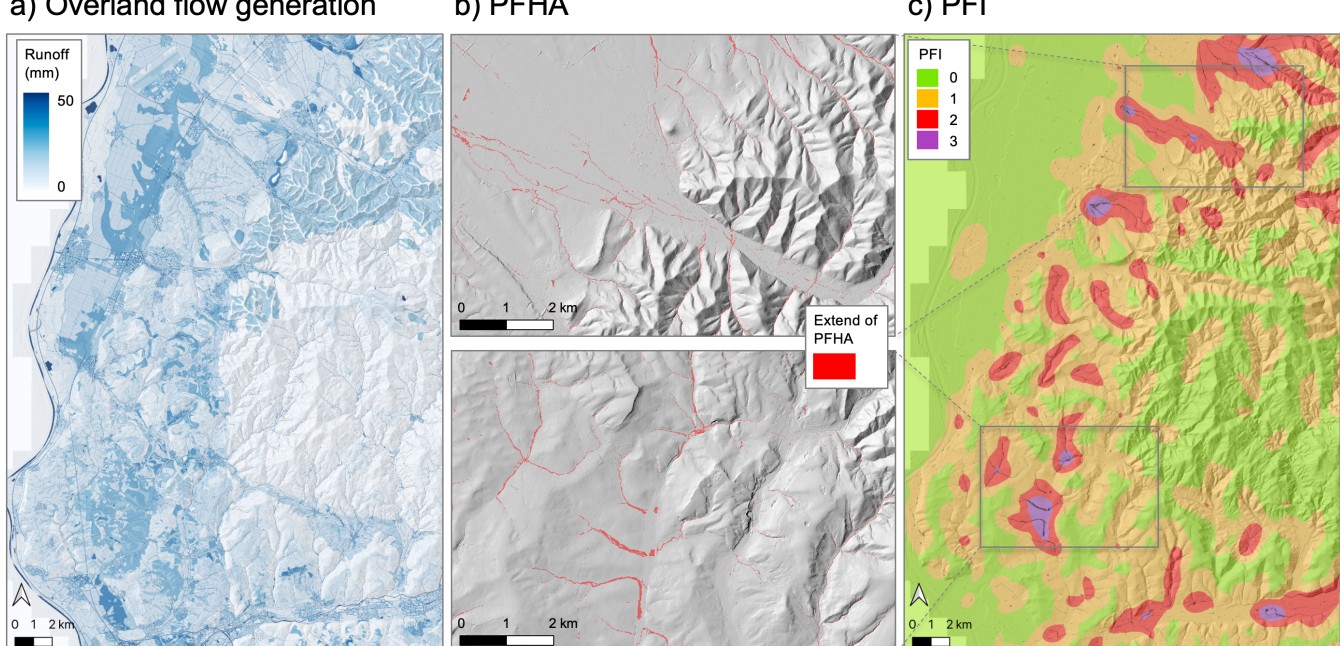

**Figure 5: The three steps to derive PFI hazard maps: a) overland flow generation for a 1h precipitation event with a return period of 100 years simulated with a hydrological model, b) the resulting PFHA in two selected areas using a 2D hydraulic model with an accumulation threshold of 10 km² and c) the resulting PFI using a moving 2 km² circular reference area. Spatial resolution is 5 m.**

Direct validation of the PFI hazard maps is challenging due to the absence of long-term data of pluvial floods. The only quantitative data accessible is the radar-derived mapping of heavy precipitation occurrences from the DWD in the CatRaRe database (Lengfeld et al., 2021). We picked events from Version 2024.01 spanning the years 2001 to 2023, focusing on occurrences exceeding the 5-year return periods determined from RADKLIM-RW. Figure 6a depicts the selected rainfall events against the overland flow generation in the background. As anticipated for events with a low return period during an

observation span of 22 years, the events plot relatively regularly and exhibit no discernible spatial pattern. Furthermore, we successfully augmented an existing database concerning the locations of flash flood incidents that caused damage to structures and roads, which was compiled by the state authorities in Baden-Wurttemberg (LUBW, 2016). The database has been updated to encompass the period from 1995 to 2024. All flash flood incidents are juxtaposed with the PFI in Figure 6b. To evaluate the impact of different accumulation thresholds on replicating the PFHA, we contrasted the original configuration of 10 km² with

a revised configuration of 20 km². The geographical distribution of flash flood incidents during the past 30 years correlates well with areas of elevated PFI, especially when investigating the PFI with a 20 km² accumulation threshold. All observed flash flood incidents occur in regions classified as PFI 2 and 3. Moreover, it is highly relevant that no occurrences were recorded in regions with a PFI of 0, although several intense rain events were recorded in these regions. It is evident that several extreme rainfall events occur in regions with a low PFI and no observation of flash floods (see below). Therefore, the

hydrological processes with the specific spatial pattern of overland flow generation and/or the terrain inhibits significant effects on the accumulation of surface runoff, resulting in the absence of simulated PFHA.



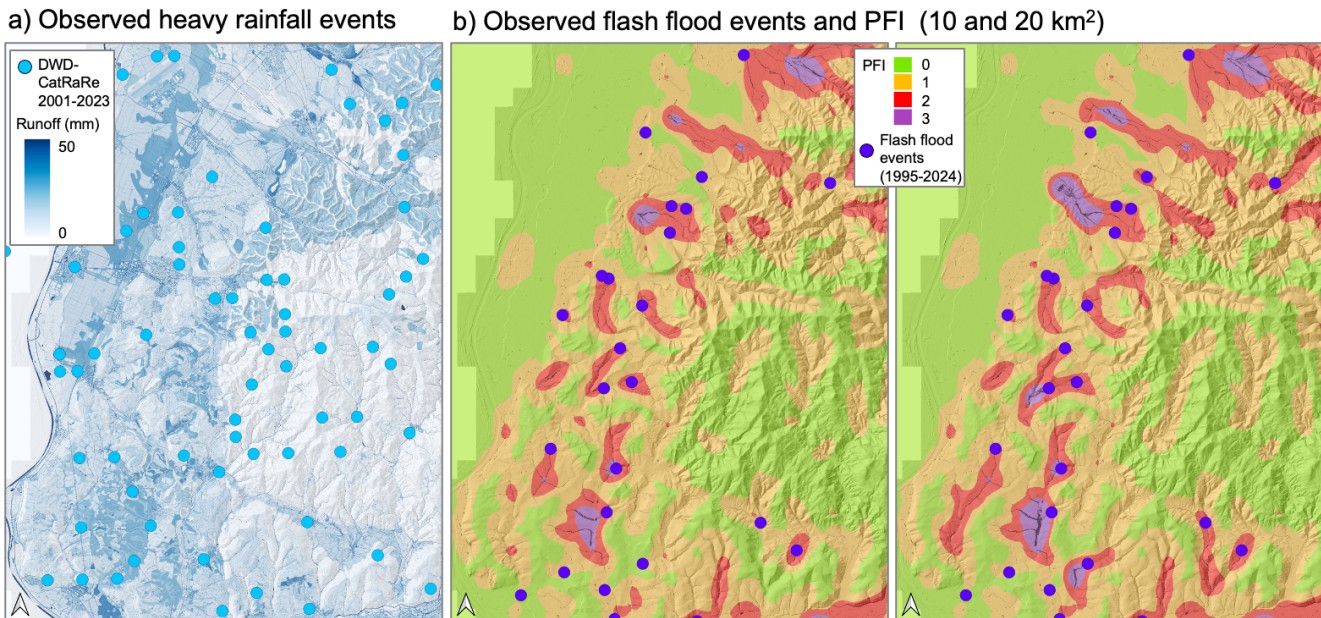

**Figure 6: Comparison of the PFI hazard map with a) observations of heavy precipitations events with a duration less than 4 hours from the DWD-CatRaRe database and b) the locations of flash floods in comparison with the PFI with an accumulation threshold of (left) 10 km² and (right) 20 km².**

## 4 Discussion

The Pluvial Flood Index (PFI) was developed as a straightforward, dimensionless index primarily for public communication. The aim is to provide information regarding pluvial flash flood hazards for early warning and the evaluation of flash flood hazards, extending beyond a simple heavy rainfall alert. To guarantee that the PFI, as a classified index, possesses substantial relevance and that relevant flash flood scenarios are precisely evaluated utilizing the PFI, the Pluvial Flood Hazard Area (PFHA) employed for the PFI classification is of paramount importance. The suggested thresholds for calculating the PFI from the PFHA fraction (Table 1) needs further evaluation. Nonetheless, the comparisons and instances in this study, along with other ongoing assessments of more historical occurrences (Krumm et al., 2025), demonstrate a somewhat solid and functional classification.

The PFI shares similarities with existing flood or landslide warning approaches that utilize rainfall thresholds (e.g., DWD, MeteoSwiss) or those that integrate rainfall thresholds with soil moisture indices (e.g., Brigandi et al., 2017). However, it directly evaluates the extent of pluvial flooding and is hence more spatially explicit. Other spatially explicit methodologies, such as the flood hazard index (FHI) (Kazakis et al., 2015; Kabenge et al., 2017) or the Flash-Flood Potential Index (FFPI) (Popa et al., 2019), rely solely on various static causal elements or Neural-Network Model and are considerably more



challenging to assess and implement in flood forecasting systems. The PFI is more advanced, as it encompasses all important hydrological and hydrodynamic processes that contribute to flash floods. However, it requires a model chain consisting of rainfall forecasts (seen as a warning system), a hydrological model and a hydrodynamic model. Given that several current fluvial flood forecasting systems already include the initial two systems, future emphasis will be on establishing a suitable

connection with hydrodynamic models or alternative methodologies that facilitate the prediction of PFHA through the generation of overland flow. Nonetheless, the hydrological models need to consider the relevant runoff generation processes in an adequate spatial resolution. Numerous federal states possess appropriate time-efficient models for area-disaggregated runoff generation calculations that account for infiltration excess, such as the state-wide RoGeR model in Baden-Württemberg (Steinbrich et al. 2021). The hydrological model LARSIM, which is already widespread for operational flood-forecasting is

an appropriate alternative (Bremicker et al. 2013). LARSIM includes an adequate dynamic infiltration module and it has proven to perform well for pluvial floods, and similar to RoGeR (Haag et al. 2022b). LARSIM models with a dynamic infiltration module are available for the German federal states of Baden-Württemberg, Rhineland-Palatinate, Hesse, North Rhine-Westphalia and parts of Bavaria as well as for Luxembourg and the French part of the Rhine catchment (Haag et al. 2022b).

The PFI can serve as a fundamental approach for operational, large-scale flash flood warning systems in the future. Such systems necessitate the effective simulation of the PFI across extensive regions. In addition to the existing hydrological models, there is a need for large-scale, computationally efficient models for runoff concentration and 2D-hydrodynamic simulations. Indeed, they present a challenge for operational calculations; however, recent advancements in high-resolution 2D hydrodynamic modelling (Khosh et al., 2024; Apel et al., 2024; Apel et al., 2022; Buttinger-Kreuzhuber et al., 2022) and rapid

alternative methods such as AccRo (Leistert et al., 2025) are now available.

Beside operational pluvial flood forecasting and warning, the PFI can also be used for extensive evaluations of pluvial flash flood susceptibility. This extensive comparative evaluation facilitates the prioritization of precautionary measures specific to pluvial flooding, including the development of comprehensive pluvial flood hazard maps at the municipal or village level (refer to the existing methodology in Baden-Württemberg) and the formulation of alarm and deployment strategies for flash

floods. The PFI hazard map presented can be readily expanded to encompass larger regions, provided that a suitable hydrological model has been established and assessed. In this context, hydrodynamic simulations pose fewer challenges, as they are required solely for particular scenarios, thereby mitigating concerns regarding the rapid execution of such models.

Recently, several approaches have been published that appear to be similar to the proposed PFI hazard maps. In Germany, the "Hinweiskarte Starkregengefahren" (https://gdz.bkg.bund.de/index.php/default/wms-hinweiskarte-starkregengefahren-wms-

starkregen.html) displays the maximum inundation depth based on a detailed hydrodynamic model for a 100 year return period and an extreme event using a high resolution DGM (1 m) and land cover data. However, it completely ignores the hydrological processes by assuming a constant runoff coefficient of 100% regardless of soil, land cover and land use or initial conditions. The spatial extent of the assumed rainfall event is also unknown, as the presented inundation maps cover large areas and overlap with fluvial flood hazard maps. As a result, they clearly do not represent pluvial flooding, but rather demonstrate the





potential of a fast 2D hydrodynamic model without taking into account the processes and scales relevant to pluvial floods. The Swiss approach "Gefährdungskarte Oberflächenabfluss" (Kipfer et al. 2018) is more sophisticated because it takes into account certain relevant hydrological processes as well as an appropriate hydrodynamic model. They assume that the simulated surface runoff "disappears" into the first water body (blue line of rivers and lakes), so their results are highly dependent on the definition of a water body and are not as consistent as the PFI's proposed fixed accumulation threshold. The national German and Swiss approaches only consider different classes of inundation depths when communicating pluvial flood hazard. We believe that the PFHA's proposed approach to defining areas where pedestrians or vehicles are at risk when water depth, flow velocity, or a combination of the two exceeds the defined thresholds is much easier to communicate and more straightforward.

Further evaluation of the approach is required, taking into account a wide range of events, locations, and options for comparing with damage data. The preliminary comparisons appear promising (Krumm et al., 2025), but more events and a comparable event database are required to improve the evaluation. We need nationwide databases of historical flash flood events. The initial compilation by Gaume et al. (2009) for Europe triggered many approaches and analyses (for example, the HiOS database of Kaiser et al., 2021), but certain event documentations are frequently lacking, such as information about damages, local observed rainfall intensities and amounts, or flooding extent; or the entire database is not publicly available due to data restrictions imposed by the insurance companies. Other online tools focus on meteorological occurrences of heavy rainfall and are less likely to provide a focused indication of pluvial floods, but they may serve as a starting point for further investigation into whether a heavy rainfall event triggered a pluvial flood or not.

Currently, the PFI only addresses pluvial hazards related to water depth and flow velocity, without other negative effects of pluvial floods such as erosion and sedimentation. Sediment has become increasingly important in fluvial flood analysis and modelling in recent years (for example, see Hamidifar et al., 2024). New approaches have been presented for detecting the damages of previous pluvial floods using satellite images, primarily due to changes in surface erosion or sedimentation (Cerbelaud et al., 2023). However, no hydro-sedimentary modelling system (e.g., Kourgialas & Karatzas, 2014) has been used for larger-scale pluvial flood real-time warning or hazard mapping to date.

## 5 Conclusion

Pluvial floods are flash floods resulting from localized, severe convective precipitation events. Climate change is expected to intensify the hydrological cycle, leading to increased heavy rainfall events and pluvial flood occurrence. The escalating threat of pluvial floods, exacerbated by urbanization and surface sealing, suggests a significant vulnerability to damage. The new Pluvial Flood Index (PFI) is a versatile tool to address these hazards and improve forecasting and alerts. The PFI considers all hydrological and hydraulic factors and is a straightforward, comprehensible, and tangible measure. Pluvial floods are rare, local events that complicate the establishment of dependable return intervals. We believe that the absolute extent of the pluvial flood hazard area (PFHA) is more crucial than the occurrence probability of an event. Therefore, the PFI is based on the relative area fraction of the PFHA to a reference area, providing a comprehensible and highly flexible basis for assessing flash flood



hazards. The dimensional index of the PFI is intended for a broad assessment of flash flood hazard, primarily for simple communication with the general public, but provides also detailed technical information about the extend and location of the PFHA. The PFHA and the PFI can serve as a fundamental approach for operational, scale-bridging flash flood warning
systems, requiring effective hydrological and hydraulic simulations across large regions. It can also be used for extensive evaluations of pluvial flash flood susceptibility, facilitating the prioritization of precautionary measures specific to pluvial flooding.

**Author contribution**

MW designed the study, and all authors carried it out and developed the PFI together. AS prepared the GIS data, HL and AH
developed the model code and performed the simulations. MW prepared the manuscript, with contributions from all authors.

**Competing interests**

The authors declare that they have no conflict of interest.

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

Index (PFI) and its application to recent inundation events in four federal states in Germany. Hydrologie und Wasserwirtschaft, in review.

Kunstmann, H., Fröhle, P., Hattermann, F.F., Marx, A., Smiatek, G., Wanger, C. (2023). Wasserhaushalt im Klimawandel. In: Brasseur, G.P., Jacob, D., Schuck-Zöller, S. (eds) Klimawandel in Deutschland. Springer Spektrum, Berlin, Heidelberg. https://doi.org/10.1007/978-3-662-66696-8_16

Leistert, H., Hänsler, A., Schmit, M, Steinbrich, A., Weiler, M. (2025) AccRo – ein recheneffizientes, quasi-hydraulisches Modell zur Ermittlung von gefährdeten Gebieten bei Sturzfluten. *Forum für Hydrologie und Wasserbewirtschaftung*, 46.25.

Landtag von Baden-Württemberg (2024) Schäden durch Hochwasser und Starkregen im Rems-Murr-Kreis, Kleine Anfrage, Drucksache 17/6981.

Lange, J., Leibundgut, C., & Simmer, I. (2003). Surface runoff and sediment dynamics in arid and semi-arid regions. *International contributions to hydrogeology*, *23*, 115-115.



Lengfeld, K., Walawender, E., Winterrath, T., & Becker, A. (2021). CatRaRE: A Catalogue of radar-based heavy rainfall events in Germany derived from 20 years of data; CatRaRE: A Catalogue of radar-based heavy rainfall events in Germany derived from 20 years of data. *Meteorologische Zeitschrift*, *30*(6), 469-487.

Lengfeld, K., Winterrath, T., Junghänel, T., Hafer, M., Becker, A. (2019). Characteristic spatial extent of hourly and daily precipitation events in Germany derived from 16 years of radar data. *Meteorol. Zeitschrift* (Contrib. Atm. Sci.), Vol. 28, No. 5, 363–378.

Lindsay, J. B., & Dhun, K. (2015). Modelling surface drainage patterns in altered landscapes using LiDAR. *International Journal of Geographical Information Science*, *29*(3), 397-411.

Liu, X. (2008). Airborne LiDAR for DEM generation: some critical issues. *Progress in physical geography*, *32*(1), 31-49.

LUBW (2016). Leitfaden Kommunales Starkregenrisikomanagement in Baden-Württemberg. 60p., ISBN: 978-3-88251-391-2, www.lubw.baden-wuerttemberg.de/wasser/starkregen.

Martínez-Gomariz, E., Gómez, M., Russo, B. (2016): Experimental study of the stability of pedestrians exposed to urban pluvial flooding. *Natural hazards*, 82(2), 1259-1278. DOI: 10.1007/s11069-016-2242-z

Martínez-Gomariz, E., Gómez, M., Russo, B., Djordjević, S. (2018): Stability criteria for flooded vehicles: A state-of-the-art review. *Journal of Flood Risk Management*, 11, 817-826. DOI: 10.1111/jfr3.12262

Oberle, P., Kron, A., Kerlin, T., Nestmann, F., & Ruiz Rodriguez, E. (2021). Diskussionsbeitrag zur Fließwider-standsparametrisierung zur Simulation der Oberflächenabflüsse bei Starkregen. *Wasserwirtschaft*, 111(4), 12-21.

Popa, M. C., Peptenatu, D., Drăghici, C. C., & Diaconu, D. C. (2019). Flood Hazard Mapping Using the Flood and Flash-500 Flood Potential Index in the Buzău River Catchment, Romania. Water, 11(10), 2116. https://doi.org/10.3390/w11102116

Rawls, W. J., Ahuja, L. R., Brakensiek, D. L., & Shirmohammadi, A. (1992). *Infiltration and soil water movement* (pp. 5-1). Handbook of Hydrology, Chapter 5.

Ries, F., Kirn, L., Weiler, M. (2020): Experimentelle Untersuchung der Abflussbildung bei Starkregen. *Hydrologie & Wasserbewirtschaftung*, 64, (5), 221-236. DOI: 10.5675/HyWa_2020.5_1

Saksena, S., & Merwade, V. (2015). Incorporating the effect of DEM resolution and accuracy for improved flood inundation mapping. *Journal of Hydrology*, *530*, 180-194.

Seibert, S. P., & Auerswald, K. (2020). *Hochwasserminderung im ländlichen Raum: ein Handbuch zur quantitativen Planung* (p. 236). Springer Nature.

Steinbrich, A., Leistert, H., Weiler M. (2016). Model-based quantification of runoff generation processes at high spatial and 510 temporal resolution. *Environ. Earth Sci.* (2016)75, 1423. DOI: 10.1007/s12665-016-6234-9.

Steinbrich, A., Leistert, H., Weiler, M. (2021). RoGeR – ein bodenhydrologisches Modell für die Beantwortung einer Vielzahl hydrologischer Fragen. In: *Korrespondenz Wasserwirtschaft* 14. DOI: 10.3243/kwe2021.02.004

Stewart, R. D., Bhaskar, A. S., Parolari, A. J., Herrmann, D. L., Jian, J., Schifman, L. A., & Shuster, W. D. (2019). An analytical approach to ascertain saturation-excess versus infiltration-excess overland flow in urban and reference landscapes.
Hydrological Processes, 33(26), 3349-3



Tarboton, D. G. (2003). Rainfall Runoff Processes. Civil and Environmental Engineering Faculty Publications, Utah State University. Paper 2570. https://digitalcommons.usu.edu/cee_facpub/2570

Vojtek, M. (2023). Indicator-based approach for fluvial flood risk assessment at municipal level in Slovakia. Scientific Reports, 13(1), 5014.

Wasko, C., Nathan, R., Stein, L., & O'Shea, D. (2021). Evidence of shorter more extreme rainfalls and increased flood variability under climate change. Journal of Hydrology, 603, 126994.

Zhang, X., Zhu, J., Wendroth, O., Matocha, C., & Edwards, D. (2019). Effect of macroporosity on pedotransfer function estimates at the field scale. *Vadose Zone Journal*, 18(1), 1-15.