# Peer review of "The Pluvial Flood Index (PFI): a new instrument for evaluating flash flood hazards and facilitating real-time warning"

_EGUsphere, 2025_

## Referee Comment (RC1)

**Review egusphere-2025-1519**

**General comments**

The authors introduce a novel index, the pluvial flood index (PFI), designed to assess and communicate the hazard potential of an area with respect to pluvial flooding. The PFI is depends on the results hydrological and hydrodynamic simulations. It increases with the fraction of a reference area where thresholds for at least one of the following variables are exceeded: inundation depth, flow velocity, or specific surface runoff. The thresholds are chosen to represent safety for pedestrians and cars. Finally the fraction is classified into four classes from "low" to "very large" hazard. This design is based on the idea that the index should be easily communicated to the public. The authors suggest to use the PFI for hazard forecasting and for the creation of hazard maps.

Especially in times of climate change it is highly important to improve disaster risk management in regard to pluvial floods and I agree with the necessity to improve existing concepts. However, in my opinion, the manuscript sometimes fails to submit to the reader the distinction of the novel aspect of the PFI and the needed models which are technically exchangeable and already existing. PFI and the underlying models can be viewed completely separately. The novel aspect of this study is solely the use of three thresholds and the moving circular window as a reference area.

Generally I wonder, if safety for pedestrians and cars should be the main indicator for pluvial flood hazard, because another main aspect of flood hazard is the damage on houses and infrastructure, which the authors do not mention and discuss.

Some parts of the manuscript describe accurately the fundamental hydrological processes that have to be represented in the models for a sound hazard estimation of pluvial floods, while other relevant aspects of the computation PFI lack some explanation. The PFI is sensitive to the parameters of the chosen circular buffer radius and the accumulation threshold. This part is missing in the "Discussion". The parameter "accumulation threshold" is never explained.

The manuscript is well written, the structure could be improved in some parts. After addressing following questions and points of concern I recommend this manuscript for publication.

I will refer to these previous points in the following detailed comments:

**Specific comments**

**Line 53:** You are suggesting the PFI as part of forecast model chain. I assume that the bottleneck are the underlying hydrodynamic models. Are there hydrodynamic models that run as part of a forecast chain on a federal state level? How long is the lead time? Could you clarify this?

**Line 70-105 ff, Relevant Processes:** This part accurately describes the processes that a model needs to represent to compute the two variables needed for the computation of the PFI (flow velocity and indundation depth). As the models are technically interchangeable and already existent, I wonder if such a detailed description of processes is necessary. Maybe this section could be shortened.

**Line 101 ff.:** Does the model you use consider geomorphological processes? Please clarify.

**Line 112:** "A hazard is defined ..." Should it not be "We define a hazard ..."? Otherwise please give a reference.

**Line 113:** Why did you base the PFI solely on safety for pedestrians and cars? In situation where vehicles float I could imagine graver impacts such as flooded houses and basements with a possible threat to life. Or are these two measures solely an indicator for the overall hazard? Furthermore, I wonder if there could even be a situation where cars are swimming while pedestrians and houses are still safe? Why this differentiation and focus on cars? Otherwise this section could be also shortened.

**Figure 2:** These two figures violate almost all principles for scientific presentations: the two figures are too small, the font size should be increased, they are not color blind friendly and the legend for the different symbols is missing, the grid is plotted over the annotations. A lot of information is missing in the caption (e.g. yellow box). This figure opens more questions than it answers and is containing information that is eventually not needed. Generally, this Figure is more confusing than helping, I think. Since you decided to just use to fixed values as thresholds for the PFI, you could think about dropping this low-quality figure entirely.

**Line 135:** Why do you use the unit $m^3 / (m * s)$ and not just use the reduced unit $m^2/s$? Where do the extra meters come from? Same for Fig 4c).

**Line 161:** You are saying that radar QPE *can* be used as an input, but at the same time you refer previously to the small spatial scale of the triggering precipitation events. In my opinion, the model results and the index can *only* be meaningful if high resolution rainfall data is used as input, which is up to now only possible with weather radar because of the low density of the rain gauge network. This should be pointed out explicitly.

**Line 165:** "weighted by radius" This becomes clear in Fig. 3c) but you should explain this a little more here by adding something like "weighted by the radius of the circular buffer to focus more on the areas close to the buffer center (see Fig. 3c)" or something similar.

**Line 178:** "based on experiences" Can you give some examples or insights from your experience here? My gut feeling also agrees with the proposed values but maybe you should just delete this statement or give some more concrete examples. See following comment to line 194.

**Line 184-187:** "To ensure ...." This could go to the discussion.

**Line 194:** This refers to my general comment, that some essential aspects of the PFI are not explained. I agree, that defining thresholds is always, at least a bit, arbitrary. However, "defined as accurately as possible" is not a sufficient description. What was the workflow when you defined it as accurately as possible?
Maybe you could already explain here why you chose these thresholds instead of just doing it at the beginning of the "Discussion" section.
Additionally I wonder, if the thresholds are a bit low: A 2 km² circular buffer has a a diameter of ~ 1.6 km. On the edges areas would be classified as hazardous because of a PFHA more than a kilometer away. In regard to communication and safety it is surely better to pick a larger than a too small area of the circular buffer to conceal uncertainties, which can be difficult to communicate. What was your motivation behind deciding for 2 km²? Why not simply use the PFHA?

**Line 222:** AccRo: You are mentioning this model various times but the reference Leistert et al. 2025 is insufficient. I could not find anything. Can you give a DOI, or a URL? Same in line 340.

**Line 232:** Radar QPE: Why is there no description about the product in the "Data" section? The rainfall product has a major impact on the outcome of your hazard assessment. To my knowledge there is no radar product with a spatial resolution of 250m in Germany. Or is it a composite product?

**Fig. 4:** a) The fonts of the rain gauge labels is a bit too small.

e) Unit of the specific surface runoff. See comment to line 135.

Subplot top right: remove label "Fig 4", little grey box not explained in caption but much later in text.

g) "resulting extent PFHA" should be labeled with f)

**Line 259:** You mention the accumulation threshold here for the first time. Do you refer to the accumulation threshold your underlying model uses to define streams and rivers? Figure 6 shows that the PFI is sensitive to this parameter. Please clarify what you mean with accumulation threshold and discuss why you chose the values you chose.

**Line 261:** The PFI is using thresholds that are aimed at the safety of humans and cars so it is not surprising that damages on infrastructure are not as well reflected by the PFI. However, this is a major impact and and the focus on cars and pedestrians slightly diminishes the informativeness of the PFI. Maybe you should also discuss this.

**Line 269:** Please provide an English translation for the non-german readers.

**Line 270 ff.:** Because of the small extent of the rainfall events which can trigger a pluvial flood, I wonder how well these small and rare events are represented in the underlying statistics of a 100-yr event. Your approach using events from CatRaRE seems to make much more sense. Maybe you could discuss this, because the rainfall input is, besides the PFI, the other major factor for the creation of a flood hazard map.

**Line 321 ff:** "The PFI is more advanced, as it …" I would change this sentence because the PFI could be calculated with any kind of velocity or inundation maps, regardless of the quality of these maps. Then the PFI would not be very reliable and would not encompass all relevant aspects. Here a clear distinction should be made: the PFI is one thing, the underlying models are something else. You can still compute the PFI based on unsuited models and then it will have little informative value.

**Line 335:** "The PFI can serve as a fundamental approach…" Relating to the previous point I find this again a bit "high-pitched". The foundation for a large-scale warning system would be a reliable model, not the PFI. The PFI is "only" aggregating the grids resulting from the models.

**Line 349:** Please translate German terms to English. Usually you would give a reference and add the URL in the references.

**Line 348-359:** This description of the "State-of-the-art" belongs to the introduction in my opinion.

**Line 359:** Please describe in more depth, which accumulation threshold you are referring to.

**Line 382:** "The PFI considers all hydrological and hydraulic factors…" Again, I think that this formulation is misleading and refer to my previous comments to line 321 and 335.

**Technical comments:**

**Line 148:** "capture the hazards of a̶ flash floods"

**Fig. 5:** b) should be "E/extent"

**Line 402:** Reference "Apel …" occurs twice.

**Line 420:** Reference format error.

**Line 437:** Reference format error.

---

## Author Comment (AC1)

**Authors' responses to the comments of Reviewer #1**

We appreciate your review and comments on our manuscript, "**The Pluvial Flood Index (PFI): a new instrument for evaluating flash flood hazards and facilitating real-time warning**". Your feedback is valuable to us, and we will make the recommended revisions accordingly. We provide detailed responses to each of your comments below.

General comments
The authors introduce a novel index, the pluvial flood index (PFI), designed to assess and communicate the hazard potential of an area with respect to pluvial flooding. The PFI is depends on the results hydrological and hydrodynamic simulations. It increases with the fraction of a reference area where thresholds for at least one of the following variables are exceeded: inundation depth, flow velocity, or specific surface runoff. The thresholds are chosen to represent safety for pedestrians and cars. Finally, the fraction is classified into four classes from "low" to "very large" hazard. This design is based on the idea that the index should be easily communicated to the public. The authors suggest to use the PFI for hazard forecasting and for the creation of hazard maps. Especially in times of climate change it is highly important to improve disaster risk management in regard to pluvial floods and I agree with the necessity to improve existing concepts. However, in my opinion, the manuscript sometimes fails to submit to the reader the distinction of the novel aspect of the PFI and the needed models which are technically exchangeable and already existing. PFI and the underlying models can be viewed completely separately. The novel aspect of this study is solely the use of three thresholds and the moving circular window as a reference area.

We agree that part of the publication deals with the definition of the PFI and its calculations and definition of the thresholds. However, another very relevant aspect of the paper is the definition of the pluvial flood hazard areas (PFHA), a combination of the three most relevant hazards (water depth, velocity and specific discharge).

Generally I wonder, if safety for pedestrians and cars should be the main indicator for pluvial flood hazard, because another main aspect of flood hazard is the damage on houses and infrastructure, which the authors do not mention and discuss.

Damage of houses and infrastructure is certainly another highly relevant flood hazard, however, the loss of lives when pedestrians are being downed or captured in vehicles should have first priority. However, the defined thresholds for water depth and velocity match well the thresholds for damage data of buildings that has been derived and analyzed in a parallel project (e.g. Singh et al., (2025) for private households or Guntu et al., (2025) for commercial buildings.

Some parts of the manuscript describe accurately the fundamental hydrological processes that have to be represented in the models for a sound hazard estimation of pluvial floods, while other relevant aspects of the computation PFI lack some explanation. The PFI is sensitive to the parameters of the chosen circular buffer radius and the accumulation threshold. This part is missing in the "Discussion". The parameter "accumulation threshold" is never explained.

We addressed these points further below in the specific comments.

The manuscript is well written, the structure could be improved in some parts.

After addressing following questions and points of concern I recommend this manuscript for publication.

The feedback to the other points is given in the specific comments below

I will refer to these previous points in the following detailed comments:

**Specific comments**
Line 53: You are suggesting the PFI as part of forecast model chain. I assume that the bottleneck are the underlying hydrodynamic models. Are there hydrodynamic models that run as part of a forecast chain on a federal state level? How long is the lead time? Could you clarify this?

At the moment this is not the case, at least in Germany. But several studies have shown the potential of directly merging hydrological and hydrodynamic model to improve forecasting of pluvial and fluvial floods and due to the rapid improvements of calculation times of hydrodynamic models (e.g. GPU), the lead time to run these models for specific areas where the hydrological forecast exceeds a certain threshold of overland flow, the lead times should be in the order of couple of minutes.

Line 70-105 ff, Relevant Processes: This part accurately describes the processes that a model needs to represent to compute the two variables needed for the computation of the PFI (flow velocity and indundation depth). As the models are technically interchangeable and already existent, I wonder if such a detailed description of processes is necessary. Maybe this section could be shortened.

We wanted to make sure that all readers, independently if they are hydrologists or not, are on the same level. However, the sections could certainly be streamlined.

Line 101 ff.: Does the model you use consider geomorphological processes? Please clarify.
No, it does not as nearly all models used for pluvial floods. But we wanted to address the general need for such models.

Line 112: "A hazard is defined …" Should it not be "We define a hazard …"? Otherwise please give a reference.
Yes, will be changed.

Line 113: Why did you base the PFI solely on safety for pedestrians and cars? In situation where vehicles float I could imagine graver impacts such as flooded houses and basements with a possible threat to life. Or are these two measures solely an indicator for the overall hazard? Furthermore, I wonder if there could even be a situation where cars are swimming while pedestrians and houses are still safe? Why this differentiation and focus on cars? Otherwise this section could be also shortened.
See also above. The PFHA are defined in order to include both measures, the water depth is more sensitive for floating of vehicles, the velocity and specific discharge more related to safety of pedestrians. We will make this clear in a revised version. To accurately predict the flooding of houses requires detailed knowledge about the structure and building of the houses (windows, garage, etc), which is usually not available at the larger scale. Hence, these details cannot be included in the PFHA but the overall defined threshold match well the damages of houses for observed pluvial events.

Figure 2: These two figures violate almost all principles for scientific presentations: the two figures are too small, the font size should be increased, they are not color blind friendly and the legend for the different symbols is missing, the grid is plotted over the annotations. A lot of information is missing in the caption (e.g. yellow box). This figure opens more questions than it answers and is containing information that is eventually not needed. Generally, this Figure is more confusing than helping, I think. Since you decided to just use to fixed values as thresholds for the PFI, you could think about dropping this low-quality figure entirely.

We absolutely agree. However, we build the figure based on already existing figures (Martínez-Gomariz et al. 2016; 2018), which already violate the principles. Therefore, we tried to improve the figures by replacing some labels and increasing the fonts, but this was not done for all, as the figure is not available as a vector file. We will try to get the original data and figure form the authors of the two publications to improve our figure, but the problems rely unfortunately in the publication in the past.

Line 135: Why do you use the unit m³ / (m * s) and not just use the reduced unit m²/s? Where do the extra meters come from? Same for Fig 4c).

We used this unit to make it clear to the reader, the is refers to specific discharge that has to be divided by the cross-sectional width. Knowing my students, I would assume that they are unable to understand the reduced units directly, so I thought it is helpful. But we could derive it once and then use the reduced units afterwards.

Line 161: You are saying that radar QPE can be used as an input, but at the same time you refer previously to the small spatial scale of the triggering precipitation events. In my opinion, the model results and the index can only be meaningful if high resolution rainfall data is used as input, which is up to now only possible with weather radar because of the low density of the rain gauge network. This should be pointed out explicitly.

Maybe we should have defined QPE in this context. We were referring to weather radar products that are improved with rain gauge data and high resolution weather models (resulting in spatial resolutions of 500-1000m and spatial resolution of 5 min) as it is already operationally in use at meteoswiss or currently under development and the German Weather Service and probably at many more-

Line 165: "weighted by radius" This becomes clear in Fig. 3c) but you should explain this a little more here by adding something like "weighted by the radius of the circular buffer to focus more on the areas close to the buffer center (see Fig. 3c)" or something similar.

Yes, will be extended.

Line 178: "based on experiences" Can you give some examples or insights from your experience here? My gut feeling also agrees with the proposed values but maybe you should just delete this statement or give some more concrete examples. See following comment to line 194.

More details form various example of already existing heavy rainfall hazard maps in the state of Baden-Württemberg can be provided to make this more concrete.

Line 184-187: "To ensure …." This could go to the discussion.

Yes, will be changed.

Line 194: This refers to my general comment, that some essential aspects of the PFI are not explained. I agree, that defining thresholds is always, at least a bit, arbitrary. However, "defined as accurately as possible" is not a sufficient description. What was

the workflow when you defined it as accurately as possible?
Maybe you could already explain here why you chose these thresholds instead of just doing it at the beginning of the "Discussion" section.
Good suggestions – we will do.

Additionally I wonder, if the thresholds are a bit low: A 2 km² circular buffer has a diameter of ~ 1.6 km. On the edges areas would be classified as hazardous because of a PFHA more than a kilometer away. In regard to communication and safety it is surely better to pick a larger than a too small area of the circular buffer to conceal uncertainties, which can be difficult to communicate. What was your motivation behind deciding for 2 km²? Why not simply use the PFHA?
The motivation was to find a compromise between your thoughts and the thoughts of reviewer #2 suggesting a much smaller area to avoid showing higher PFI on hillslopes. In general, the presented approach to visualize the PFHA is one possibility to do so and we will make it clearer, that there are certainly several other approaches, as for example directly mapping the PFHA (as we also do – see example Figure 4). However, the PFHA cannot be seen when mapping large areas, so either the PFI could be calculated based on other units (catchments, political boundaries, etc.), but as discussed, this would always change the PFI as the related area is different. Therefore, we developed our approach in close cooperation with many potential users and optimized in it several stakeholder workshops – but we will put a bit more emphasize on the decisions and why it was developed like presented in the paper.

Line 222: AccRo: You are mentioning this model various times but the reference Leistert et al. 2025 is insufficient. I could not find anything. Can you give a DOI, or a URL? Same in line 340.
We are in the process of preparing a publication for the model – hopefully it will be available as pre-print beginning of September 2025. Meanwhile we will provide a link for a conference contribution: https://uni-freiburg.de/unr-hydro/wp-content/uploads/sites/94/406-TdH-2025-Leistert-Haensler-Schmit-Steinbrich-Weiler_dinA4-300dpi.pdf

Line 232: Radar QPE: Why is there no description about the product in the "Data" section? The rainfall product has a major impact on the outcome of your hazard assessment. To my knowledge there is no radar product with a spatial resolution of 250m in Germany. Or is it a composite product?
We will add a brief description about the Radar product in section 2. We used the data from the Kachelmann Group, which provides QPE in 250m x 250m resolution for recent years (since 2020)

Fig. 4: a) The fonts of the rain gauge labels is a bit too small.
We will adapt the font size
e) Unit of the specific surface runoff. See comment to line 135.
Subplot top right: remove label "Fig 4", little grey box not explained in caption but much later in text.
We will remove the label and explain the box.
g) "resulting extent PFHA" should be labeled with f)
We will adapt this

Line 259: You mention the accumulation threshold here for the first time. Do you refer to the accumulation threshold your underlying model uses to define streams and rivers? Figure 6 shows that the PFI is sensitive to this parameter. Please clarify what you mean with accumulation threshold and discuss why you chose the values you chose.

Thank you for pointing this out. We will include a section on the accumulation threshold earlier when we explain the concept of PFHA and PFI. The idea behind the threshold is that when using design storm events as input (in our case a 100yr event), we have widespread heavy rainfall input which would lead to unrealistically high flooding in the downstream areas. In reality, convective rainfall events have a certain spatial extent only, somehow limiting the amount of water available for pluvial flooding. In order to mimic this, a maximum accumulation area is introduced after which it is assumed that the capacity of the river is large enough to capture the accumulated flow.

Line 261: The PFI is using thresholds that are aimed at the safety of humans and cars so it is not surprising that damages on infrastructure are not as well reflected by the PFI. However, this is a major impact and the focus on cars and pedestrians slightly diminishes the informativeness of the PFI. Maybe you should also discuss this.
See also above. We will discuss this more in detail.

Line 269: Please provide an English translation for the non-german readers.
Yes, will be provided

Line 270 ff.: Because of the small extent of the rainfall events which can trigger a pluvial flood, I wonder how well these small and rare events are represented in the underlying statistics of a 100-yr event. Your approach using events from CatRaRE seems to make much more sense. Maybe you could discuss this, because the rainfall input is, besides the PFI, the other major factor for the creation of a flood hazard map.
Thank you for pointing this out, so we can make this clearer. Actually there are two different approaches we aim for in the application of the PFI. First is that we calculate the PFI for real events – also to show the potential for forecasting. In this case we would use radar QPE (like they are reflected in CatRaRe). Second case is the general susceptibility for pluvial floods. In this case we need to work with design rainfall events. The 100yr design rainfall event we use is based on the extreme value analysis of 1 hour precipitation events recorded at stations and then spatially interpolated. In this case the data is available for the full state of BaWü at a spatial resolution of 1x1km and was the standard data for the pluvial risk management in the state of BaWü in the first version. Alternatively, one could use the KOSTRA dataset (5x5 km resolution).
The comparison with CatRaRe in the figure was mainly to highlight the fact that there are certainly much more heavy rainfall events all over the place then we had recorded pluvial flood events. We will make this clearer in the manuscript.

Lne 321 ff: "The PFI is more advanced, as it …" I would change this sentence because the PFI could be calculated with any kind of velocity or inundation maps, regardless of the quality of these maps. Then the PFI would not be very reliable and would not encompass all relevant aspects. Here a clear distinction should be made: the PFI is one thing, the underlying models are something else. You can still compute the PFI based on unsuited models and then it will have little informative value.

You are certainly right. We will adapt the section.

Line 335: "The PFI can serve as a fundamental approach…" Relating to the previous point I find this again a bit "high-pitched". The foundation for a large-scale warning system would be a reliable model, not the PFI. The PFI is "only" aggregating the grids resulting from the models.
Again, you are certainly right. We will adapt the section pointing out the need for reliable model simulations.

But still, the PFI - based on its aggregation concept can provide a large scale pluvial flood warning not yet present.

Line 349: Please translate German terms to English. Usually you would give a reference and add the URL in the references.
Yes, will be accordingly

Line 348-359: This description of the "State-of-the-art" belongs to the introduction in my opinion. Line 359: Please describe in more depth, which accumulation threshold you are referring to.
We will move the description of the 'Hinweiskarte Starkregengefahren' as well as the example from Switzerland to the introduction but keep the comparison with the PFI in the discussion

Line 382: "The PFI considers all hydrological and hydraulic factors…" Again, I think that this formulation is misleading and refer to my previous comments to line 321 and 335.
We will reformulate the sentence.

Technical comments:
Thank you for pointing them out. We will correct them.
Line 148: "capture the hazards of a flash floods"
Fig. 5: b) should be "E/extent"
Line 402: Reference "Apel …" occurs twice.
Line 420: Reference format error.
Line 437: Reference format error.

---

## Author Comment (AC2)

**Authors' responses to the comments of Reviewer #2**

We appreciate your review and comments on our manuscript, "**The Pluvial Flood Index (PFI): a new instrument for evaluating flash flood hazards and facilitating real-time warning**". Your feedback is valuable to us, and we will make the recommended revisions accordingly. We provide detailed responses to each of your comments below.

This article introduces a novel approach to assessing and mapping pluvial (flash) flood risk, referred to as the Pluvial Flood Index (PFI). The proposed method is demonstrated through two case studies: first, a hindcast of a flash flood that occurred on June 2, 2024, in the Wieslauf catchment in Germany; and second, the generation of flood hazard maps for a region of approximately 1,000 km² in the southwest of Baden-Württemberg. In the latter case, the reliability of the PFI maps is evaluated using a database of flash flood incidents that caused damage to infrastructure and roads. This database, originally compiled by the state authorities of Baden-Württemberg and extended by the authors, covers events from 1995 to 2024.

The proposed approach, along with its underlying motivation and evaluation, may initially appear appealing; however, upon closer examination, it raises several concerns.

First, the method does not actually generate new information but rather involves a post-processing of standard runoff maps produced using a combination of a rainfall-runoff model and a 2D hydraulic model. This post-processing is carried out in two steps. In the first step, a Boolean variable called PFHA (Pluvial Flood Hazard Area) is assigned to each pixel based on predefined flood depth and velocity thresholds, categorizing them as either low or high hazard. In the second step, a spatial smoothing technique is applied to calculate, for each pixel, a Pluvial Flood Index (PFI), which reflects the proportion of high-hazard pixels within a circular buffer surrounding that pixel. Four proportion ranges are defined, corresponding to four levels of PFI. However, this spatial smoothing—central to the PFI concept—is not adequately justified, and the choice of the circular buffer size (2 km²) remains unexplained.

It is understandable that the initial motivation behind the development of the PFI method was to enhance the readability of flood risk maps at a large spatial scale. In the two case studies presented, the "Pluvial Flood Hazard Areas" (PFHA) are mostly confined to riverbeds and broader floodplains, making them difficult to discern on large-scale maps. To address this, the proposed PFI method uses spatial smoothing to create broader, more visually prominent zones around clusters of high-hazard PFHA pixels. While this improves map legibility, it is crucial to recognize that spatial smoothing is a digital artifice. The resulting PFI levels are not solely the product of hydrological and hydraulic modeling—they are shaped by additional processing choices and assumptions, and should therefore not be overinterpreted.

Unfortunately, the authors appear to let their enthusiasm override critical assessment. One must ask: is there any sound reason to believe that hillslopes located in the surroundings of inundated floodplains are more exposed to pluvial flooding than hillslopes located elsewhere? A careless reading of the PFI map might suggest so, but such a conclusion lacks physical basis. Proximity to a floodplain does not inherently increase a hillslope's vulnerability to pluvial flooding.

In short, the PFI method should be presented for what it truly is: a straightforward graphical tool designed to highlight clusters of high flood hazard at the regional scale, not a physically grounded index of flood exposure.

Thank you for this critical assessment of the general PFHA and PFI methodology. We agree that the PFHA and PFI could be considered as a simple post processing of the output of a hydrological and 2D-hydraulic model chain and could be applied to any rasterized, large-scale data of maximum water depth, maximum flow velocity and maximum specific discharge – if these variables are available.

First, we will improve to show the location of the PFHA. As we did not include the location of rivers in Fig 5a, it looks like that PFHA are only confided to riverbeds and broader floodplains. But this is not the case for many regions. I have included an example below, and we will provide a similar map and information in a revised paper to showcase the different locations of PFHA

[Figure]

One point we will make clearer in the revised version of the manuscript is the spatial scale the PFI aims at. As we stated in 177ff, the PFI is intended for a broad assessment of pluvial floods at a regional (e.g. state-level ~ 200x200km) scale. It should allow decision makers to get a quick information if a certain community or region is generally prone to pluvial flooding (in the case of flood hazard maps) or if a certain rainfall event has the potential to trigger a hazardous flood event in a certain community/region (in the case of forecasting). Hence the PFI is not designed for distinguishing the flood vulnerability of single hillslopes or local features or buildings. This information shall be taken from the detailed PFHA maps or already existing local pluvial flood maps, but the PFI can provide the information which pluvial flood map is the most relevant in the current situation.

We will restructure section 2 in a way that the general purpose of the PFHA and the PFI and the link to already existing pluvial flood maps becomes more clear. We will also discuss the pro and cons of the different methods for spatial aggregation (e.g. could also be on political/organizational units) more in detail and detail more why we think that the chosen approach (with radial weighting) has some benefits compared to the others.

Regarding the underlying hydrological and hydraulic processes, we still think that this is relevant to be included in the manuscript, since a meaningful PFHA and related PFI (for both, operational forecasting and identifying general pluvial flood susceptibility) requires the sound modelling of hydrological and hydraulic process. As already discussed, these processes are often not represented in current developments (e.g. 'Hinweiskarte Starkregen') implemented at state or even federal level. To make it clearer that this is not a direct part of the PFI but a prerequisite, we would suggest to move the chapter 2.1 outside the definition of the PFI.

Second, the evaluation procedure is insufficiently described and critically discussed. In Figure 6, past damaging flood events are depicted as dots. However, flood events have spatial extents—especially those occurring in catchments with upstream drainage areas of 10 to 20 km². How was the representative location of each flood event determined for the evaluation? It appears likely that the dots indicate the locations where the most significant damages occurred, typically along riverbeds and, in many cases, within flood-prone plains. If this is the case, then the apparent skill of the flood risk assessment may not stem from the PFI method itself, but rather from the underlying rainfall-runoff and hydraulic simulation models that generate the base maps.

We agree that it would be beneficial to have better damage data to verify the PFI. However, there is hardly any information available in the case of pluvial flood events at the scale of interest. For a few villages, we have some more explicit spatial information, but this is typically related only to the location of emergency calls during the flooding. The database which is available for the state of BW and included in Fig. 6 is more or less a binary data set, indicating if in a certain community there was a damage following a heavy rainfall event or not. Hence the dots simply represent communities where there was a damage. One other possibility could be to mark the whole area of the village, but depending on these boundaries, this may also be misleading. But we will make this clearer in the text.

Since the quality of the PFHA/PFI depends on the quality of the models used to estimate the parameters (maximum water depth, maximum flow velocity and maximum specific discharge) to define hazard areas, it is simply impossible to disentangle the quality of the PFI from the quality of the underlying model data. We will also make this clearer in the revised version.

For the Wieslauf example, we collected some data on maximum inundation – but of course only for a few locations and a few days after the catastrophic event. Although this data does not at all allow the identify all spots affected by the pluvial flood, it could be included to verify that PFHA/PFI structures somehow are reflecting inundated areas.

Moreover, although the article is ostensibly focused on pluvial flooding, the evaluation is confined to areas with catchment sizes larger than 10 to 20 km², which more closely align with riverine or flash flood events rather than true pluvial floods. This choice contradicts the stated objective of the manuscript. What proportion of the DWD-CatRaRe database was excluded due to this threshold? It would be helpful to visualize all flood events from the database in Figure 6, using a distinct color to differentiate those included in the evaluation. Additionally, the relationship between the recorded heavy precipitation events (Figure 6a) and the damaging flood events (Figure 6b) is unclear—some commentary on their correspondence is needed.

It seems that this is a misunderstanding which probably arises from the fact that we did not describe the accumulation threshold and its motivation in sufficient detail. The PFI aims at large scale, however, in order to reflect the spatial scale of convective precipitation events and the typical catchment areas related to pluvial floods, we limit the accumulation area of generated flows to 10km² and 20km², respectively.

So we do not consider any catchment areas larger than 10 to 20km², depending on the chosen threshold. Hence we also did not exclude any convective, short and small scale events from CatRaRe, but the opposite – we excluded all events with a duration of more than 4h, which can be assumed to be rather of frontal nature. We also did not exclude events from the 'damage' database, but used all data that was reported – definitely knowing that it is incomplete.

The comparison between heavy precipitation events (Figure 6a) and the damaging flood events (Figure 6b) was included to show that there were many more heavy rain events (actually more or less all over the study region as can be expected) compared to the damage events (which of course are somehow linked to the valleys or hillslopes – however not to all valleys) – so basically this shows the added value of a PFI forecast with respect to a solely precipitation forecast. We will reformulate the section so that this becomes clearer.

Further concerns arise in Section 2.1, "Relevant Processes." The section implies that pluvial runoff and flash floods are primarily caused by direct overland flow due to either saturation or infiltration excess. This is an outdated and overly simplistic view that has long been challenged in hydrological literature. Decades of research into hillslope processes and flash flood generation have demonstrated the complexity and spatial-temporal variability of runoff mechanisms, including the often-dominant role of subsurface flows (interflows). Such simplification is not appropriate for publication in an international journal and should be reconsidered. Since detailed discussion of rainfall-runoff processes is not central to the manuscript's main contribution, it may be best to remove or significantly revise this section.

We agree that interflow can play an important role but mainly for events with longer duration. For short events under typically dries soil moisture conditions, interflow is quite unlikely in our regions, which we can show with a larger body of observational data. Therefore, we wanted to make the case of the relevance of saturation or infiltration excess overland flow for the generation of pluvial floods. Numerous sprinkling experiments revealed the importance of excess infiltration in the case of short-term high-intensity heavy rainfall events representing the main cause for pluvial flooding (e.g. Ries, F., Kirn, L., & Weiler, M. (2020). Runoff reaction from extreme rainfall events on natural hillslopes: a data set from 132 large-scale sprinkling experiments in south-western Germany. *Earth System Science Data*, *12*(1), 245-255).

As mentioned previously we think that the correct representation of the dominant runoff generation processes is key for a meaningful estimation of the PFI. Hence, we suggest to leave the section in, streamline it, but also restructure the method part of the paper.

In conclusion, the article presents a potentially useful approach. However, the PFI method should not be overstated. It should be honestly described for what it is: a simple graphical tool aimed at highlighting clusters of flood hazard on regional maps. If the primary focus is on pluvial flooding, then there is no justification for excluding events in small upstream catchments—especially when past records of such events exist. While including these areas may lead to less favorable evaluation outcomes, such results would be highly relevant and valuable for the community working on pluvial flood risk mapping. If the current threshold is retained, then the manuscript should clearly state that its focus lies with riverine or flash floods rather than pluvial floods.

As mentioned above, we will restructure the manuscript to make it clearer what the PFI method is based on and for what it designed, but also what the prerequisites are to have a sound estimate of pluvial floods. We will reformulate section 3.2 so it becomes more clear that we did not exclude any small upstream catchments (the focus was on explicitly including them!) from the analysis and focus more on the need of defining accumulation thresholds.

Finally, the discussion and analysis need to be further developed. It appears that the PFI and PFHA indices primarily identify flood-prone river plains. If this is indeed the case, it should be explicitly acknowledged and critically considered.

We disagree that PFHA are primarily linked to river plains (see response above), but some valley (with and without rivers!) are certainly included, which of course makes sense, since water usually accumulates not at hill tops. However, if one examines the spatial PFI patterns in Fig 6 closely, one can see that not all river-plains are characterized per see as high hazard areas but that regional differences due to runoff generation processes as well as the 2d hydrodynamic flow accumulation are present. If a simple topographical index would be used, the regional differences would be much less pronounced than in the case of the PFI. The same would be true, if we would not consider the runoff generation processes but simply use rainfall estimates as input for the 2d-hydrodynamic flow accumulation – which as already discussed, are still represented in many current developments (e.g. 'Hinweiskarte Starkregen'). We will reformulate the text to make it clearer.